# The Perception and Roles of School Mental Health Professionals Regarding School Bullying (*Suluk Audwani*) in Oman: A Qualitative Study in an Urban Setting

**DOI:** 10.3390/ijerph21080991

**Published:** 2024-07-28

**Authors:** Muna Al Saadoon, Rawaa Abubakr Abuelgassim Eltayib, Amjad Hassan Alhaj, Moon Fai Chan, Said Aldhafri, Samir Al-Adawi

**Affiliations:** 1Department of Child Health, College of Medicine & Health Sciences, Sultan Qaboos University, Muscat 123, Oman; munasa@squ.edu.om; 2Department of Family Medicine & Public Health, College of Medicine & Health Sciences, Sultan Qaboos University, Muscat 123, Oman; s136163@student.squ.edu.om (R.A.A.E.); moonf@squ.edu.om (M.F.C.); 3Sociology and Social Work, College of Arts and Social Sciences, Sultan Qaboos University, Muscat 123, Oman; amjadalhaj@squ.edu.om; 4Department of Psychology, College of Education, Sultan Qaboos University, Muscat 123, Oman; aldhafri@squ.edu.om; 5Department of Behavioral Medicine, College of Medicine & Health Sciences, Sultan Qaboos University, Muscat 123, Oman

**Keywords:** bullying, Omani schools, focus group discussions, school mental health programs, perception of bullying, linguistic contexts, *Suluk Audwani*

## Abstract

With increasing mental health risks among school populations and prevalent bullying, school mental health professionals (SMHP) are crucial globally. This study explores the perspectives of SMHP on bullying in Omani schools, focussing on definitions, types, current practices, and future strategies. Involving 50 Omani SMHP from Muscat Governorate with at least three years of experience, data was collected through structured interviews and analyzed using Braun and Clarke’s six-step thematic analysis. Six key themes emerged: The definition of bullying, its components, common types, current anti-bullying practices, challenges, and future suggestions. Bullying, termed “سلوكعدواني” (*Suluk Audwani*), meaning aggressive behavior, includes five components: perpetrators, victims, harmful behavior, spectators, and psychosocial factors. Verbal bullying, physical bullying, and cyberbullying are common and vary by age. SMHP frequently employ awareness-raising and psychological first aid. Challenges include resistance from students and parents and institutional barriers. In the future, SMHP will emphasize greater awareness to effectively address *Suluk Audwani.* Oman has adopted international best practices to recruit SMHP. SMHP’s perspectives on *Suluk Audwani* reflect both their training and Omani cultural influences. Future research should explore various social strata to improve evidence-based understanding and prevention of *Suluk Audwani*.

## 1. Introduction

Bullying in schools is a global public health problem with significant short- and long-term physical, psychological, and social effects on victims and perpetrators [1,2,3]. Given its widespread impact, it is crucial to understand how various social groups perceive and address school bullying behavior. Studies have shown that various members of society, including teachers, school administrators, social workers, law enforcement officers, and school nurses, play a key role in protecting children’s well-being. It is increasingly recognized that school bullying can lead to poor mental health outcomes. With increasing mental health risks among the school population and the prevalence of bullying [4], school mental health professionals (SMHP) are increasingly used in educational settings around the world [4,5], and professionals under the SMHP rubric are increasingly used in educational settings in various parts of the world [5]. Following consultations with international and regional experts, the Eastern Mediterranean Regional Office of the World Health Organization (WHO EMRO) developed a school mental health program aimed at addressing behavioral and socio-emotional issues in students [6]. Oman, a country in the Arabian Gulf, has also seen an increase in the employment of school mental health programs in school settings [7]. 

Oman, with its unique Islamic heritage and maritime history [8], presents different social dynamics. Existing studies have focused on the prevalence and characteristics of bullying [9,10,11,12,13], but there is a lack of qualitative research exploring the perspectives of SMHP. To our knowledge, to date, no qualitative investigations have been conducted that attempt to understand the bullying issue from the perspective of the Omani society. The population is pyramidal in structure and consists mainly of children and adolescents. The preponderance of children and adolescents in school has resulted in large class sizes and all the consequences that this may entail. Although there is a proliferation of private schools, most of the children in Oman attend universally free government schools [14]. Little has been explored about the perception and attitude towards school bullying among SMHP in Oman. Studies have indicated that different areas of schools tend to have different perspectives on the problem of bullying in their schools. To date, the perspective of SMHP has received little attention. 

In the academic context discourse, most of the SMHP education is largely derived from the Euro-American conception of human nature due to the hegemony of the Anglo-American curriculum [6,15]. Therefore, it is fair to generalize that the perspectives of SMHP are significantly shaped by their Western education in the humanities and psychology, which emphasizes individual autonomy and self-actualization, which, in turn, has pervaded core ethical principles, advocacy, and ethical decision-making [16]. However, it is not clear how this background among SMHP is handled in diverse cultural backgrounds such as Oman. In addition to education, culture influences our values, beliefs, norms, and symbols, which, in turn, guide our actions and decisions. Therefore, it would be important to highlight some of the cultural factors that could explain how bullying behavior is perceived. First, studies have shown that cultural norms significantly influence behaviors considered bullying. Seminal studies by the anthropologist Edward T. Hal hypothesized that society could be dichotomized into a “high-context culture” versus a “low-context culture” [17]. These two modes represent different ends of a continuum in communication styles within cultures. According to this hypothesis, high-context cultures rely more on non-verbal cues and subtle gestures for communication, while low-context cultures emphasize direct verbal communication. It has been speculated that Arab Gulf countries tend to fall into high-context cultures where indirect communication and honor preservation are highly valued, potentially affecting the way bullying is perceived and addressed [18]. In addition to such communication styles, Arab culture is more likely to lean towards interdependence, culminating in a collective mindset [16]. Historically, Arab societies were organized around tribes and extended families, creating a strong sense of community and mutual support. This tribal structure emphasized loyalty, cooperation, and collective responsibility. It has not yet been established how this sociocultural pattern influences the perception of bullying among SMHP. 

There is a scarcity of qualitative studies that examine the perspectives and experiences of professionals who deal with bullying behaviors. Most existing studies have focused mainly on describing and interpreting the experiences of bullies, victims, and onlookers themselves [19], but little attention has been paid to SMHP. Therefore, this study aimed to gain an in-depth understanding of bullying behaviors and antibullying practices in Omani schools from the perspective of SMHP. First, this study seeks to understand how SMHP defines and narrates the bullying topology that they encounter in schools in Oman. Second, the study aims to decipher current anti-bullying practices that SMHP spreads in schools. An integral part of the present quest is to gather what SMHP envisions for future practices to effectively address school bullying within the sociocultural context of Oman.

## 2. Materials and Methods

### 2.1. Setting

This study constitutes part of a larger study known as “Culturally Sensitive Quantification and Strategies for Intervention to Counter School Bullying in Muscat Governorate”. Being a national strategic study, it draws the cooperation from the Ministry of Education (MoE), a single body that oversees the government and private schools in Oman. To lay the foundations for this study, the contact persons of the Ministry of Education organized a symposium with representatives of SMHP, as well as other sectors of the MoE. The symposium was designed to highlight the objective of the study with its counterpart in the MoE. The second objective was to use the presence of SMHP at the symposium to explore their perspectives on bullying behaviors in Omani schools, focusing on how they define and perceive bullying, the common types encountered, current anti-bullying practices, and suggested future strategies. Contact persons were explicitly informed to include representatives of SMHP from government schools in the Muscat Governorate, which consists of six provinces (*wilayats*), Al Amarat, Bawshar, Muscat (old town), Muttrah, Qurayyat, and Al Seeb. This setting was chosen due to its diverse and dense population, which provides a comprehensive view of school bullying in an urban context.

### 2.2. Study Participants & Recruitment

The target population for this qualitative study was school mental health professionals (SMHP) in Muscat governorate, Oman. Initially, the Ministry of Education (MoE) was contacted about this research and its purpose. After discussions with the MoE, invitations were sent to registered SMHP currently working in the Muscat Governate who also satisfied the inclusion criteria. Overall, of those who were invited, 50 mental health professionals from various schools agreed to participate in this study. 

What constitutes SMHP was adopted from a manual known as the Directorate General of Primary Health Care & Department of School & University Health [7]. What is defined as SMHP are graduates of education, humanity, and social sciences, with qualifications in educational psychology and educational social work [20]. 

#### Inclusion/Exclusion Criteria

The inclusion criteria for this study were Omani, were registered SMHP as defined by the Directorate General of Primary Health Care & Department of School & University Health [7], had at least three years of experience as a mental health professional in schools in one of the six provinces of the Muscat governorate, had witnessed school bullying, and had subjectively confirmed that they were knowledgeable about school bullying. 

### 2.3. Data Collection

This study was conducted on 22 October 2023. Focus group discussions using structured interviews were the main methodology for the data collection in this study. Morgan & Hoffman [21] state that some things must be considered when conducting focus groups to ensure meaningful interactions and high-quality data collection. First, it is recommended to pay attention to the composition of the group. Therefore, for this study, a homogeneous group composition was chosen, all groups comprising a combination of SMHP. Second, for the present purpose, the focus group size consisted of 7–8 participants for each group. The present number of participants appeared to echo other studies [22,23,24]. Similarly, the study used the saturation phenomenon, by which researchers continue to collect data until the response content appears to be saturated [21,25].

### 2.4. Procedure

Consenting participants were invited to a lecture room, where an introductory statement was given on the general scope of the study. The participants then received a demographic survey to gather basic information (such as years of experience, the schools in which they work and the province in which they belong). After the introductory session, the consenting participants were divided into groups, and the focus group discussions began.

To preserve maximum comfort and attention of participants during their participation in the discussions, the research questions were split into two sessions. In addition, to further this goal, the participants were given a 45-min breakfast break between sessions. Both sessions were held on the same day. Each focus group session ranged from 1 h and 30 min to 1 h and 45 min. The present moderators were assigned the role of asking pre-planned questions to obtain the objectives of the study, refocus the discussion if it digressed, and facilitate the development of the relationships of the participants. The closed-ended question item was divided into two sessions, as shown in Table 1. This was taken to accommodate fatigue and to be relevant to the theme. The first theme was comprehensively characterizing bullying, and the second theme focused on preventive and treatment strategies that they employ in their school setting. The guide questions used by the moderators were derived after a lengthy discussion between present researchers and information drawn from the corpus of literature [26,27,28,29].

The focus group discussions were held in a cloistered private room. Data were collected from the groups until saturation was achieved. After six groups, it was decided that no more novel information could be acquired from the groups, and thus, saturation was achieved in this investigation.

### 2.5. Recording Procedure

Conventionally, voice recorders are used to document the narration of the participants. However, many participants felt uncomfortable with recording. Therefore, the paper-and-pencil format was used, in which respondents received blank papers to fill in their answers to each question asked of them. According to best practices [24], the participants of the focus group interview received a briefing on the study objectives and had the opportunity to raise related concerns or questions. 

### 2.6. Data Analysis

This study used thematic analysis as a basis for qualitative data analysis. Mainly, it followed the six-step process outlined by Braun and Clarke [30] for thematic analysis. First, become familiar with the data itself. Second, generate preliminary and systematic codes as you go through the data. Third, re-organize the codes into tentative themes. Fourth, review the themes created. Fifth, finalize the final set of themes and/or subthemes for the research. Finally, write the findings around the chosen themes/subthemes.

Initially, the handwritten narration of the participants was converted into Microsoft Word text. A researcher (R.A.A.E.) entered the Arabic responses into a qualitative data analysis software ATLAS.ti^TM^ (version 24.1) for Windows [31], where codes, themes, and sub-themes were generated. Subsequently, the same researcher, who is bilingual, translated the research findings and quotes from Arabic to English. Three other researchers (S.A., S.D. and A.A.) reviewed the codes and themes generated, which will be described in the Results section.

### 2.7. Rigor

To ensure the scientific merit of the study, the research team took several measures. After a researcher (R.A.A.E.) translated the themes and quotes into English, three other bilingual researchers reviewed the translations (M.S., S.D. and A.A.) to ensure that the requested information was not misrepresented due to linguistic differences between Arabic and English. Furthermore, two other researchers (S.A. and M.C.) who are fluent in the English language further reviewed the English interpretation of the themes/subthemes.

### 2.8. Ethical Considerations

Ethical approval was sought from the local IRB of the Medical Research Ethics Committee of the College of Medicine and Health Sciences, Sultan Qaboos University (SQU-EC/004/2022, MREC#2701). Before participating in this investigation, the individuals were asked to sign an informed consent form. Furthermore, they received assurances that their data would be anonymized to protect their privacy. In addition, they were explicitly told that their participation is voluntary and that they are free to withdraw from the study at any time they deem appropriate. 

## 3. Results

### 3.1. Demographic Data

In total, 50 mental health professionals from different schools participated in six focus group discussions for this study. The majority of SMHP who participated in this study were women (60%), fell into the age group between 31 and 40 years (62%), and had between 11 and 20 years of experience (52%). Furthermore, most of the participants came from schools located in Al Seeb province (50%) and were in schools in Cycle 2 (5th to 10th Grade) (60%). For a summary of the descriptive statistics of the respondents in this study, see Table 2.

### 3.2. Thematic Analysis Results

When conducting focus group discussions with SMHP in Omani schools, several themes emerged while analyzing their responses. An overview of all the themes, sub-themes, and highlight quotes is given in Table 3.

#### 3.2.1. The First Theme: Bullying Definition

When the participants were asked to give the terms they use to define bullying and what encapsulates the behavior itself, some essential terms were highlighted. The terminology used by SMHP to describe bullying is “سلوك عدواني“ or “*Suluk Audwani*”, which roughly translates into aggressive behavior. For an overview of the terms given by the respondents during the focus group discussions, refer to Table 4.

In general, the agreed definition of bullying employed by the Omani SMHP involved in this study is “A repeated deliberate aggressive behaviour practised by an individual (or a group), with the intention of harming another individual (or a group), and there is usually a difference in power between the perpetrator and the victim. The repeated deliberate aggressive behavior can have negative consequences for the parties involved in it and the community as a whole.”

#### 3.2.2. The Second Theme: Components of Bullying

In this subtheme, the SMHP identified five key elements of bullying (or, in this instance, *Suluk Audwani*). First, they refer to the perpetrators, who could be either an individual or a group of individuals.


*“One of the essential components of bullying is the perpetrator, and they have a group (typically supporters) who are usually prepared to help the perpetrator in their bullying behaviour.”*
[G4]

Second, to make *Suluk Audwani* happen, there must be a victim, which can be one person or a group.


*“The victim is typically a weak-willed individual who cannot defend himself or herself and has little self-confidence.”*
[G4]

Third, a key element of *Suluk Audwani* is the behavior itself. It is usually a deliberate act that is carried out with the intention of physically or psychologically harming the other party.


*“The Bullying tool, technique, or bullying method.”*
[G1]

Fourth, during the occurrence of *Suluk Audwani*, there is usually a crowd of spectators that gather and witness the behavior. In particular, SMHP have highlighted that crowds can behave actively or passively. Active witnesses to bullying have been described as not being impartial. During *Suluk Audwani* observation, active spectators tend to defend the victim or inform an authority figure. On the other hand, passive crowds, which are usually how the majority of people react, prefer to remain neutral while witnessing *Suluk Audwani*.


*“A key element of bullying is the audience, who can be passive (the majority): They prefer to remain silent or withdraw when they witness bullying. Their actions encourage the perpetrator and exacerbate the problem. The crowds can also be active: They defend victims or inform the administration about the situation.”*
[G3]

Fifth, the respondents stated that certain psychosocial circumstances or traits are breeding grounds for *Suluk Audwani* incidents. Bullies, for example, are more likely to engage in *Suluk Audwani*, particularly those with certain characteristics. They tend to be physically strong, known to be attention seekers, or those who have been previously exposed to bullying. Victims, on the other hand, are more likely to be perceived as lacking confidence and having low self-esteem. They also describe their home situation as lacking support. It was also stated that victims tend to have peculiar physical traits or attributes that become prominent for others bullying them.


*“In the bully: Low academic achievement, behavioural problems, their surrounding environment (habitual use of harmful language), previous exposure to bullying, and attention seeker. In the victim: Lack of self-confidence or low self-esteem, lack of emotional support at home, and the presence of a trait or characteristic that predisposes the student to bullying.”*
[G3]


*“Strong personality, smart and resourceful student, wants attention from teacher and social worker (bully), weak personality and unable to defend itself, problems in emotional support from the family and lack of self-confidence (victim).”*
[G4]

#### 3.2.3. The Third Theme: Common Types of Bullying

The respondents in this study have reported four types of *Suluk Audwani*, which are listed in order of occurrence. First, the most common form of *Suluk Audwani* in Omani schools is verbal bullying. In this type, bullies like to laugh aloud at a trait or characteristic exhibited by the victims. 


*“One of the most prominent forms of bullying is verbal bullying, such as threatening, belittling, laughing at looks, voice, height, weight, nicknames, skin color, nationality, stuttering, their mothers and mocking the father’s workplace (or nature of work).”*
[G1]


*“Verbal bullying: Profanity, name-calling, comparison and belittling, threats, sarcasm, and name calling.”*
[G3]


*“The most prevalent form is verbal bullying: Repeated name calling, ridiculing the victim’s mother, mocking congenital abnormalities such as in speech, stuttering, physical appearance, or disability.”*
[G6]

Next, the second most common form of *Suluk Audwani* is that which falls under the rubric of physical bullying. SMHP have mentioned some examples of this type:


*“Physical bullying: touching, hitting, pushing, throwing and stealing personal items.”*
[G1]


*“Forcing victims to engage in dangerous actions [that will get them in trouble] including stealing others’ belongings.”*
[G6]

The third most common type of *Suluk Audwani* in Omani schools is cyberbullying, in which social networks serve as its main vector.


*“Popular examples of cyberbullying include taking unwanted photos of the victims, laughing at them on social media groups, posting their voice messages without their permission, making a video mocking them, commenting profanity on their social media posts, impersonating them on social media sites, and threatening to delete their account.”*
[G1]

The respondents in this study mention social bullying (or social exclusion) as the fourth most common form of *Suluk Audwani* in their schools. Social bullying is embodied by any act of exclusion that a person does to another person with the intention of keeping them without any social support. Popular examples of social exclusion given by the SMHP are


*“Social bullying, such as spreading rumours.”*
[G4]


*“Psychological bullying (exclusion): Isolation, labeling, stigmatization.”*
[G4]

In particular, race-based bullying also occurs. However, the SMHP in this study were divided on where to classify them. Some respondents mention it as a manifestation that falls under social bullying.


*“Social bullying: includes bullying that relates to an individual’s tribal heritage, traditions, or customs.”*
[G1]


*“Social or psychological bullying, such as exclusion, stigma, stares, theft, and racial discrimination.”*
[G4]

However, other SMHP in this study believe that it should have its own form and be called “ethnic bullying”. Under this label, victims are mocked for belonging to a different ethnicity or tribe.


*“Ethnic bullying: racial prejudice.”*
[G6]


*“Ethnic bullying is related to making fun of color or race.”*
[G3]

It is worth noting that the participants have indicated that sometimes bullying forms can differ according to several influencing factors, such as


*“Bullying differs somewhat between the village and the city in terms of its manifestations and influencing factors.”*
[G3]


*“Manifestations of bullying can vary according to the school structure, the state, the community, student density, age group, or gender.”*
[G6]

Furthermore, age and gender appear to play a role in the type of *Suluk Audwani*. Physical *Suluk Audwani* appears to be more common in younger students, while verbal *Suluk Audwani* is more common in older students.


*“In girls’ schools (fifth through tenth grade), verbal bullying is the most common, followed by cyberbullying and psychological bullying (exclusion). In girls’ schools (grades one through four), physical bullying is more common, followed by verbal bullying.”*
[G3]


*“In the lower grades it is mostly verbal and then turns into brawls [in the upper grades].”*
[G6]


*“Verbal bullying: It is prevalent in cycles (1–8) ... Physical and behavioural bullying: Prevalent in grades (10–12).”*
[G1]

#### 3.2.4. The Fourth Theme: Current Practices against Bullying

In this theme, SMHP discuss the preventive and mitigation measures they currently employ in their respective schools. First, to reduce the occurrence of bullying or *Suluk Audwani*, SMHP have listed a number of tactics they employ, including workshops and group counseling sessions. Some examples they mentioned are as follows:


*“Orientation classes and lectures where we talk about bullying and psychological first aid. In addition, programs to promote tolerance and friendship among students.”*
[G5]


*“Daily psychological counselling on topics of self-improvement topics, and mentoring sessions titled: ‘We are all friends’, ‘Be kind’ and ‘Self-confidence’. Also, individual or group counselling programs for students who lack confidence and self-worth, as well as for bringing in experts to present lectures on this topic.”*
[G2]


*“Counselling programs (emotional venting, relationship building, why are you changing?), individual sessions to provide students with effective communication and relationship building skills, and hosting experts to talk about the topic.”*
[G6]

In addition to addressing the issue at the institutional level, SMHP highlighted the importance of addressing *Suluk Audwani* from a structural perspective, including using all forms of media to reach the community in general.


*“Raising awareness through radio programs (to promote the concepts of friendship, tolerance, respect, kindness, and altruism), In addition, use the school’s morning assembly broadcast to spread awareness among students about the psychological and physical effects of bullying and to paint an unfavorable mental image of the bully.”*
[G5]


*“Implementing preventive awareness programs within the social worker’s duties (such as social media campaigns, radio shows, and electronic bulletins posted on school accounts), and spreading the value of tolerance, altruism, and respect within the school.”*
[G1]


*“Designing various advisory boards, radio programs (puppet theatre), lectures for students and parents about cyberbullying at the morning assembly and distribution of brochures and fliers on social media.”*
[G4]

In particular, as a key preventive measure, the respondents highlighted the importance of occupying students’ free time with multiple school-related activities.


*“Arranging extracurricular activities for pupils and motivating them to do so in order to pass the time and let off steam. In addition, hold contests and small-scale displays that showcase students’ viewpoints.”*
[G5]


*“Educational workshops that increase student awareness of the issue and clarify its effects, and skills’ workshops aimed at building healthy relationships.”*
[G6]

Among the tactics used to minimize *Suluk Audwani* in schools, the participants tended to impart behavior techniques and other coping strategies. These strategies are directed both at the perpetrator and at the victim. Consequently, perpetrators are commonly trained on how to moderate their emotions, while victims are often given psychotherapeutic sessions to cope with the sequala of *Suluk Audwani*.


*“Individual counselling for the bully, with the student learning about the dangers of bullying and correcting his/her perception of bullying and its negative effects on the bully and the bullied. [and for the victim] Engage the student in group and school activities to foster friendships and positive relationships that reduce bullying, promote speaking and communication, and increase self-confidence.”*
[G4]


*“For the bully, counselling sessions on controlling and overcoming anger and sessions on learning about rational and irrational thoughts. As for the bullied, attempt to integrate them with other students who possess integrity and establish an environment that is conducive to their needs.”*
[G3]


*“For the perpetrator, engage them in artistic and skill workshops aimed at calming and venting anger and emotions, as well as interactive participatory workshops to strengthen relationships, change perceptions, build role models, and therapeutic breathing exercises.”*
[G6]

Furthermore, the participants believe that both victims and perpetrators are implicitly and explicitly informed that on school grounds, “actions have consequences.”


*“Implement the Student Affairs Regulation as a measure to control student behaviour and conduct.”*
[G6]


*“Disciplinary actions (the student is referred to the expert handling Student Affairs Regulation): the student signs a pledge, is warned, or is dismissed (after agreement with the school administration).”*
[G4]

#### 3.2.5. The Fifth Theme: Challenges

When asked about the challenges SMHP typically face when trying to implement their planned practices against bullying, they outlined several points that can be classified largely into several domains. 

First, most of the SMHP cited that both parents and students tend to be resistant to participating in the measures applied by SMHP in issues related to bullying.


*“For example, the student’s lack of receptivity to the intervention plan (or lack of response), the student’s refusal to attend scheduled sessions, stubbornness, their lack of acceptance of change—some are not amenable to discussion and behavioural change. Furthermore, it is difficult to change the belief (self-image/mental image) of the bully and the victim.”*
[G6]


*“One of the difficulties is the victim’s fear of speaking up. In addition, school bullies are usually aggressive and challenging to deal with.”*
[G4]

Additionally, the participants noted that parents play an unhelpful role in implementing some mitigation practices. The examples of this given by the SMHP were as follows:


*“Lack of response from certain parents, who either don’t understand the seriousness of bullying and its impact on victim mental and social well-being or firmly believe that students should handle things on their own.”*
[G6]


*“Typically, parents are not open to the concept of bullying since they prefer to view it as simply another normal child’s fight. In addition, they like to overtly coddle their children and will not allow their child to be designated as a “bully.”*
[G2]


*“Some parents lack understanding of discipline (hit whoever hits you).”*
[G5]

Second, the participants noted that they face some obstacles from the administration and teaching cadre that limit them when implementing some measures.


*“Difficulties from the school administration when attempting to apply the Student Affairs Regulations.”*
[G6]


*“Lack of cooperation from management and teaching staff to solve daily problems or situations in the classroom, and shortage of teachers in schools leading to too many spare classes and increased pressure on mental health professionals in school.”*
[G4]

In addition, it is challenging for SMHP to conduct counseling sessions or uphold the confidentiality of students seeking assistance because management typically allocates them offices close to the administration offices, making students wary of visiting the social worker’s office.


*“One problem is that in certain older schools, the psychologist or social worker does not have a private space (like a private office).”*
[G5]


*“The geographical location of the social worker’s office and its presence in the administration. In addition, there are sometimes too many student patients in the specialist’s office and there is a lack of confidentiality.”*
[G5]

In addition, SMHP face some difficulties in organizing counseling sessions and classes. Some examples mentioned by them were as follows:


*“Challenges in organizing and implementing psychological counselling and group counselling, and the lack of available classes to implement these sessions in.”*
[G3]


*“Absence of appropriate spaces to hold certain programs, and issues with students refusing to leave class to go to individual or group therapy.”*
[G2]

Lastly, the participants mentioned how difficult it is for them to implement effective preventive and therapeutic initiatives due to the lack of resources.


*“Lack of financial support to implement preventive and therapeutic projects.”*
[G1]

#### 3.2.6. The Sixth Theme: Suggested Practices for the Future

To further consolidate anti-bullying measures, the participants highly recommended that this issue be directly addressed to society—for example, to distribute health education on bullying in all strata of *Suluk Audwani*. This increased public awareness is important as a preventive measure. These align with the view that “prevention is better than cure”.


*“Educate parents and students, make sure government agencies and stakeholders give the issue serious importance, and raise community awareness through: Local media, Friday prayers, and social media.”*
[G1]


*“Launching an awareness story series and an animated children’s educational series on the local children’s channel. In addition, leverage the role of Learning Resource Centres’ role in providing visuals on the topic, and encouraging government agencies or stockholders to run family awareness campaigns. It’s crucial since some cases of bullying arise from people’s emotional response to their home and domestic abuse.”*
[G4]

Furthermore, the participants suggested that concerted efforts should be made to provide in vivo awareness to children in addition to their caregivers. The existing curriculum in the country should feature bullying and its effects on society. Additionally, most of the information currently available in the country is anecdotal or impressionistic. Therefore, robust studies are needed to investigate this phenomenon in the country.


*“Training workshops that focus on cases affected by bullying for everyone involved (everyone benefits, not just the victim), and how to deal with bullying issues.”*
[G3]


*“Awareness programs implemented from the specialist’s work plan (social or psychological), including a set of educational values to deal with the school and family environment, applying programmes on tolerance: Ideally an event on Tolerance Day, and the “I Matter” initiative to modify behaviours.”*
[G5]


*“Curriculum, and then curriculum: The syllabus needs to be modified appropriately, like adding a unit on good values and qualities to instil them in students’ minds. Other suggestions include reducing the total number of subjects, increasing Islamic education classes, and assigning grades on behaviour throughout the year.”*
[G4]


*“Addition of a unit in the Islamic education curriculum that focusses on: Instilling good values and improving self-confidence.”*
[G3]


*“Establishing a specialized academy to reformulate bullies’ thoughts and examine them in a scientific, objective, and focused way. Increase in research studies on which we can rely and base our programs on.”*
[G6]

## 4. Discussion

### 4.1. Discussion of Themes

Oman has a significant proportion of its young population in the school-age group. Consistent with global trends, Oman is witnessing a rising tide of youth mental health crises and school bullying behavior [9,10,11,12]. To address this trend, the country has a protocol initiated by the Eastern Mediterranean Regional Office of the World Health Organization (WHO) to develop a school mental health program to address behavioral and socio-emotional issues in schools [6]. This initiative has led to the training of a cadre of humanities and social sciences graduates to deliver the school’s mental health program [32]. These school mental health professionals (SMHP) are increasingly addressing behavioral and socio-emotional issues in schools, including those related to bullying [7]. The perception of bullying among SMHP is crucial to understanding and addressing bullying in schools. SMHP may view bullying differently than students and parents, which can lead to discrepancies in defining and managing it. As key figures in prevention and intervention, SMHP can collaborate with parents, teachers, and other staff, which is essential for preventing and mitigating bullying in schools. 

This study aims to explore the perspectives of SMHP on bullying, focusing on its characteristics and the strategies used to prevent and address it. The qualitative study was conducted in the Muscat governorate, where a large part of the country’s population resides. The participants consisted of 50 mental health professionals from this specific catchment area. The first theme that emerged included definitions and components, common types, challenges, and suggested practices for the future. These themes will be discussed in the following and, when relevant, will be recapitulated with Omani’s sociocultural perspective, as well as within existing literature. 

#### 4.1.1. Definition of Suluk Audwani

The narrative of a participant indicated that bullying is known in Arabic as *Suluk Audwani*. According to Arabic lexicons, the word *Suluk* refers to the set of actions of an individual in response to internal and external stimuli. The second part of the term, *Audwani*, is a derivative of the word *Audwan*, which refers to an unjust attack, harm, or injustice. An ‘*Audwani* person’ is a person who assaults another with his tongue or weapon. Thus, this term encapsulates the idea of aggression toward others. Like its counterpart in international nosology, the topology of bullying also includes not just physical entities but also amorphous entities such as emotional or psychological concepts. The entrance of amorphous entities is interesting. Previous studies in the region have indicated that when misconstruing occurs in the Arabian Gulf population, individuals in such societies tend to be less inclined to report psychological or emotional symptoms [33] but rather articulate their predicament in somato-physical metaphors [34]. Therefore, the present definition improvised by SMHP in Oman appears to echo the *sine qua non* of the contemporary definition of school bullying behavior that could be a physical or non-physical entity [35,36,37,38]. 

#### 4.1.2. Components of Suluk Audwani

SMHP in Omani schools have stated that *Suluk Audwani* is characterized by repeated aggressive behavior, often with a power imbalance, causing harm to another individual or group. Components such as perpetrators, victims, specific behaviors, bystanders, and underlying psychosocial factors have emerged under the bullying characterization. Olweus [38,39] has laid the foundation for the definition of bullying, which includes three key criteria: intentional aggressive behavior, the repetition of the act, and a power imbalance between the parties involved. Although the definition of the present cohort echoes those of Olweus, there are some additional features. Consequently, bullying behavior is only legitimate if it has negative consequences in the community. In Oman, where social norms and values heavily influence behavior, bullying is perceived not just as individual harm but also as a threat to etiquette. This reflects the cultural emphasis on avoiding shame and maintaining honor [40]. Consequently, bullying can be tolerated if it reinforces social hierarchies but is condemned if it overtly causes shame [16]. This view appears to highlight the interconnection between individual actions and communal repercussions in a shame-based society like Oman [41]. Therefore, there is an indication that, in the case of the perpetrator, if they do not disrupt social harmony, bullying, in the local vernacular, is the act known as *shujaa*. The concept of “*shujaa*” in Arabic society refers to a person who is brave, courageous, and strong [41]. In the context of bullying, the term “*shujaa*” can be used to describe individuals who can stand up to bullies and protect themselves or others from harm. This concept is rooted in the cultural values of Arab societies, which emphasize the importance of courage, honor, and the protection of one’s family and community [34]. This suggests that definitions of bullying may vary depending on the specific social and cultural environment. However, veneration *shujaa* has its downsides. For example, victims are unlikely to report their situation to their parents, teachers, or SMHP. Future studies should examine these issues further. 

Within the aforementioned discussion, it would be relevant to compare perceptions of bullying and poor mental health among SMHP in Arab societies and western countries. First, it should be noted that there is a lack of studies that have specifically focused on SMHP in the region despite recommendations from the World Health Organization to improve school mental health. Despite such a caveat, the perception of bullying, albeit indirectly, could be drawn from Arabian Gulf societies and hence compared to the trend in the global north, sometimes known as the rich industrialized country of the north. As alluded in the present study, the perception of bullying, on the one hand, is influenced by international best practices and international treaties such as the Convention on the Rights of the Child [16], which assumes the universality of human psychology and, on the other hand, the situation on the ground that has little affinity with what is prescribed as international best practice [42,43]. The global north perspective emphasizes individual autonomy and psychological well-being, which frames bullying as a violation of personal rights and an infliction of mental harm. This perspective influences the robust anti-bullying policies and mental health support systems in schools. In contrast, in traditional society in the Arabian Gulf, bullying may be perceived more as a disruption to social harmony and a source of shame for both victims of perpetrators [16]. However, SMHP, through their training, are required to perceive bullying behavior to align with international best practices. Within this disjunction, the study highlights the need for customized professional development programs that address the specific cultural context of Arabian Gulf societies, as Western training programs often emphasize evidence-based psychological practices that may not fully translate to the cultural needs of SMHP in societies in the global south. Therefore, more studies are warranted on this line of research. 

#### 4.1.3. Common Types of Suluk Audwani

In addition to the definition of bullying, this qualitative study has also evaluated common types of bullying. The prevalent forms of bullying in Omani schools include verbal, physical, cyber, and social bullying, each characterized by different behaviors and manifestations. A respondent stated that verbal bullying involves insults, name-calling, and ridicule, while physical bullying encompasses actions such as hitting, pushing, and stealing. Social bullying, or social exclusion, involves isolating individuals or spreading rumors to undermine their social standing. Cyberbullying, facilitated by digital platforms, includes tactics such as online harassment and impersonation, which seems to be consistent with the trend in society. There is a high penetration of Internet services in Oman [44]. This reveals its dark side, including the proliferation of cyberbullying and screen time [45].

Our sample stated that types of bullying can vary depending on age, with verbal bullying more common in older students and physical bullying more common in younger students. The relationship of age with bullying types is a point of contention in the literature. Although some studies produced results similar to ours [46,47], others appear to dissent from the present findings [48]. Therefore, it is worth speculating about the entry of culture. In many societies, including Oman, respect for elders and hierarchical structures is deeply ingrained. Therefore, older students can exert their power through verbal means, aligning with the cultural emphasis on verbal communication and respect for authority. On the contrary, younger students, who may not have developed sophisticated verbal skills, might resort to physical aggression to assert dominance, reflecting the more primal nature of their interactions.

Ethological studies of animal behavior highlight different roles within social groups, mirroring the dominance hierarchies and aggressive behaviors observed in human societies [49]. Similarly, social psychology provides frameworks such as social identity theory and the bystander effect, elucidating how individuals’ behavior within groups is influenced by their social identities and the presence of others [50]. In this context, the present study examines the roles involved in bullying, identifying five key components: perpetrators (and their supporters), victims, spectators (both passive and active), the behavior itself, and psychosocial circumstances or traits that facilitate bullying. These roles align with the concept commonly referred to as “The Bullying Circle” by Olweus [38], which describes various roles that pupils play in bullying situations. These roles include the “bully” (the main perpetrator), “henchmen” (supporters), “victim” (targeted individual), “passive spectators” (observers who do not intervene), and “active defenders” (those who attempt to help the victim). This qualitative understanding provides information on how different individuals contribute to and are affected by bullying situations, highlighting the complexity of the social dynamics involved. 

#### 4.1.4. Current Practices against Suluk Audwani

Some of the narrations that came from SMHP regard their current practices that they employ in their workplaces in schools. For brevity, these are clustered into four issues: (i) education and awareness, (ii) support for victims, and (iii) disciplinary actions. Regarding education and awareness, in their current practices, SMHP said that whenever there is an opportunity to interact with students, they use these encounters to promote awareness of bullying. Whenever an opportunity arises, they inform them about the detrimental effects of bullying on its victims and those of its ostracizing behavior on society in general. Furthermore, when the context is right, they raise awareness that the *Suluk Audwani* is an expression of emotion, and, like many emotions, they tend to be transient. Therefore, the best emotion to cultivate is empathy, which is good for both the victim and the perpetrator. There is evidence in the literature that increasing empathy tends to be strongly associated with the attenuation of aggressive behaviors that are intimately linked to bullying. Van Noorden et al. [51] have reported a systematic review of the role of empathy in mitigating bullying behaviors. In their literature search, the authors found 40 articles that align with their inclusion criteria. The study unequivocally suggests that improving empathy has a direct impact on attenuating bullying.

Another practice highlighted by SMHP is to focus on support. First, SMHP are equipped with psychological first aid. The purpose of this program is to provide them with psychological resources that prevent them from experiencing the adverse effects of bullying. This commitment aligns with the current literature that indicates that when traumatic events occur, affected individuals receive first aid, resulting in the prevention of more intransigent ill effects of trauma [52,53], but there is a dissenting view [54]. It is worth highlighting that these psychological first aids were not administered to victims of school bullying.

The final point with regard to current SMHP practices is about what to do toward perpetrators. Oman is one of the signatory countries of the United Nations Convention on the Rights of the Child (UNCRC) [16]. These imply that what is known as corporal punishment has been abolished in Omani schools. In this sense, the current practice toward perpetrators involves signing a pledge and giving warnings if the offense is repeated, leading to a possible dismissal from the school. These warnings to perpetrators are often made in conjunction with experts in school administration and student affairs [55].

#### 4.1.5. Challenges Facing SMHP Regarding Suluk Audwani

In a society that values social modesty, the presence of school mental health programs (SMHP) on campuses may be viewed with contempt by both students and their families. In Oman, psychological distress is often attributed to supernatural forces such as jinn, evil eyes, and sorcery [56]. Therefore, what is considered psychological distress is often addressed by the traditional healing system [57]. Traditional healers offer a sense of confidentiality and cultural understanding, making them a preferred option for people who fear judgment or discrimination when seeking help from modern mental health specialists. Although psychotherapy addresses intrapersonal issues such as psychic pain or guilt, traditional healing focuses on interpersonal conflicts, aligning more closely with cultural beliefs and values [56]. 

Consistent with socio-cultural teachings, SMHP report that parents dislike their children being scrutinized, leading to a lack of cooperation from the children themselves. As a result, when approached by SMHP, children often become defensive or minimize the incident. Such responses from parents have been documented in other populations [58,59,60]. SMHP also face lukewarm support from administrative staff and a lack of resources dedicated to promoting school mental health programs.

In summary, SMHP face numerous challenges in implementing anti-bullying measures, including resistance from students and parents, administrative obstacles, logistical limitations, and resource limitations. Students often hesitate to participate in intervention programs, and parents can minimize the severity of bullying. Administrative hurdles, such as limited cooperation and physical space for counseling sessions, further hinder efforts to address bullying effectively. Additionally, financial restrictions hinder the implementation of preventive and therapeutic projects. 

#### 4.1.6. Suggested Practices for the Future for Suluk Audwani

Lastly, the respondents suggested several practices to address school bullying. First, they highlighted the importance of launching national education campaigns. Second, in conjunction with these campaigns, the school curriculum should also include issues of school bullying. Studies elsewhere have indicated that the introduction of an antibullying curriculum tends to result in a significant reduction in school bullying [61]. Third, study participants have drawn attention to the need for more thorough research to examine this phenomenon throughout the country. They indicated that more research is needed to examine the magnitude of the problem and consider the best practice to come to grips with *Suluk Audwani.* To achieve this, participants have emphasized the importance of raising awareness of bullying among students and the broader community, along with a focus on values-based education. In the literature, there are several community education programs to reduce bullying. They emphasized that a knowledgeable student body, equipped with values like empathy and tolerance and enriched social relationships, is less likely to engage in bullying behaviors. For example, Australia’s “Classroom as Community” approach reduces bullying by fostering harmonious relationships and promoting empathy, kindness, and responsibility among students [62]. Other initiatives such as “Parent and Teacher Councils,” “Digital Health Interventions” [63], and “Assertiveness Therapy” [64]. Although such an undertaking appears to constitute a wish list, it remains to be seen whether there is a strategy for improving such techniques.

Furthermore, the participants stated the need for anti-bullying initiatives not only to target students but also to involve their parents and the wider community. In Oman, where family ties and community cohesion are paramount, the participation of parents and community leaders in antibullying efforts is aligned with cultural norms and values. By engaging parents, who often play a central role in shaping children’s attitudes and behaviors, and enlisting the support of community leaders, such as religious figures or tribal elders, anti-bullying campaigns can leverage existing social structures to foster a collective commitment to combating bullying. This approach not only improves the effectiveness of prevention strategies but also reinforces the message that addressing bullying is a shared responsibility that requires the active participation of all members of society. Furthermore, by involving the larger community, antibullying initiatives in Oman have the potential to promote a culture of empathy, mutual respect, and solidarity, which are fundamental principles in Omani society.

In summary, SMHP advocates increased public awareness, educational initiatives, and robust research to combat school bullying. Engaging the community through media campaigns and religious sermons, integrating bullying awareness into the curriculum, and conducting studies to inform evidence-based interventions are recommended strategies. By addressing these challenges and implementing suggested practices, schools can create a safer and more supportive environment for all students, mitigating the prevalence and impact of bullying.

### 4.2. Limitations

This type of study is likely to be marred by various limitations. The most explicit are highlighted here. First, the discussion focuses solely on the perspectives of SMHP. Future studies could examine the views of students, parents, and teachers. Such an approach could provide a more holistic understanding of bullying in Omani schools. Taking this into account, the present convenient sample (*n* = 50) was from the urban part of Oman, Muscat Governorate, which may not be representative of all SMHP in Oman or other regions. Additionally, the phenomenon of school bullying can vary from urban to rural areas. Second, the use of a paper-and-pencil format to record responses, instead of audio recordings, may have limited the richness and depth of the data collected. Given that Arab culture traditionally values oral communication over written communication, relying on written responses may have limited the depth and expressiveness of the input of participants [65]. Third, the study involved the translation of responses from Arabic to English, which can introduce translation bias and a possible loss of meaning. Finally, the study has overlooked some of the important factors that contribute to the socioeconomic status, ethnicity, or specific school situation of school bullying. These variables are likely to be important precursors to the path of school bullying. 

## 5. Conclusions

SMHP in Omani schools have extensively characterized bullying, outlining its definition, components, common types, challenges, and suggested practices. The definition of bullying encompasses repeated deliberate aggressive behavior to harm another individual or group, often accompanied by a power imbalance. Components include perpetrators, victims, specific behaviors, bystanders, and underlying psychosocial factors. Common types of bullying include verbal, physical, cyber, and social bullying, with ethnic bullying also identified as significant, although debated. The challenges of addressing bullying range from student and parent resistance to administrative obstacles, logistical limitations, and resource limitations. Proposed practices for the future emphasize increasing public awareness through community participation, integrating anti-bullying education into the curriculum, and conducting research to inform evidence-based interventions, ultimately creating a safer and more supportive school environment for all students.

## Figures and Tables

**Table 1 ijerph-21-00991-t001:** Overview of the questions asked of the participants in the focus groups.

First Session:	
Q1	In your opinion, what is the definition of bullying?
Q2	What are the most prominent appearances of bullying you have noticed in your school? (Also give examples from your school)
Q3	How common is bullying in your school? (Additionally, mention the most common types of bullying in your school)
Q4	What are the elements of bullying?
Second session:	
Q1	What are the preventive measures you are currently practicing at your school?
Q2	What are the treatment strategies you currently employ in your school?
Q3	What obstacles do you face when trying to implement prevention or treatment initiatives in your school?
Q4	What programs would you recommend implementing to deal with bullying?

**Table 2 ijerph-21-00991-t002:** Summary of demographic data of participants (School Mental Health professionals).

Measure	n (n = 50)	%
Gender:		
Male	20	40%
Female	30	60%
Age Group:		
<=30	5	10%
31–40	31	62%
41–50	14	28%
**Years of experience:**		
0–10	21	42%
11–20	26	52%
21–30	3	6%
**Province (Wilaya):**		
Al Amarat	8	16%
Bawshar	7	14%
Muscat (Old Town)	3	6%
Muttrah	4	8%
Qurayyat	3	6%
Al Seeb	25	50%
**School level:**		
Basic Education: Cycle 1 (Grades 1–4)	4	8%
Basic Education: Cycle 2 (Grades 5–10)	30	60%
Post-Basic Education/Secondary (grades 11–12)	10	20%
Combined Basic and Post-Basic School	6	12%

**Table 3 ijerph-21-00991-t003:** Summary of the main themes, subthemes, and main quotes of this study.

Themes	Main Quotes
First Theme: Bullying Definition	“سلوك عدواني” or “*Suluk Audwani”* “A repeated, deliberate, aggressive behaviour is practised by an individual (or a group), with the intent to harm another person (or a group), and there is usually a difference in power between the perpetrator and the victim. The repeated deliberate aggressive behavior can have negative consequences for the parties involved and the community as a whole.”
Second Theme: Components of Bullying	“One of the essential components of bullying is the perpetrator, and they have a group (typically supporters) who are usually prepared to assist the perpetrator in their bullying behaviour.” [G4] “The victim is usually a weak-willed individual who is unable to defend themselves and has little self-confidence” [G4] “The Bullying tool, technique, or bullying method” [G1] “A key element of bullying is the audience, who can be passive (the majority): They prefer to remain silent or withdraw when they witness bullying. Their actions encourage the perpetrator and exacerbate the issue. The crowds can also be active: They either defend victims or inform the administration about the situation.” [G3] “Strong personality, smart and resourceful student, wants attention from the teacher and social worker (the bully), weak personality and unable to defend himself, emotional support issues from the family, and lack of self-confidence (the victim).” [G4]
Third Theme: Common bullying types	“In girls’ schools (fifth through tenth grade), verbal bullying is most common, followed by cyberbullying and psychological bullying (exclusion). In girls’ schools (grades one through four), physical bullying is more common, followed by verbal bullying.” [G3]
Fourth Theme: Current Practices	“Orientation classes and lectures: Talk about bullying, psychological first aid, and implementing programmes to promote tolerance and friendship between students.” [G5] “Raising awareness through radio programs (to promote the concepts of friendship, tolerance, respect, kindness, and altruism), In addition, use the school’s morning assembly broadcast to spread awareness among students about the psychological and physical effects of bullying and to paint an unfavourable mental image of the bully.” [G5] “For the bully, counselling sessions on controlling and overcoming anger and sessions on learning about rational and irrational thoughts. As for the bullied, attempt to integrate them with other students who are trustworthy and establish an environment that is conducive to their needs.” [G3] “Disciplinary actions (the student is referred to the expert handling Student Affairs Regulation): the student signs a pledge, is warned, or is dismissed (after agreement with the school administration).” [G4]
Fifth Theme: Challenges	“For example, the student’s lack of receptivity to the intervention plan (or lack of response)” [G6] “Typically, parents are not open to the concept of bullying since they prefer to view it as simply another normal fight between children. In addition, they like to overtly coddle their children and will not allow their child to be designated as a “bully.” [G2] “Difficulties from the school administration when attempting to apply the Student Affairs Regulations.” [G6] “Challenges in organizing and implementing psychological counselling and group counselling, and the lack of available classes to implement these sessions in.” [G3] ‘Lack of financial support to implement preventive and therapeutic projects.” [G1]
Sixth Theme: Suggested practices for the future	“Educate parents and students, make sure government agencies and stakeholders give the issue serious importance, and raise community awareness through: Local media, Friday prayers, and social media.” [G1] “Addition of a unit in the Islamic education curriculum that focusses on: Instilling good values and improving self-confidence.” [G3] “Establishing a specialized academy to reformulate bullies’ thoughts and examine them in a scientific, objective, and focused way. Increase in research studies on which we can rely and base our programs on.” [G6]

**Table 4 ijerph-21-00991-t004:** List of the most common Arabic terms used by Omani School Mental Health Professionals to describe bullying and their English translation.

Arabic Terms	Transliteration	English Translation	*n* (%)
**“** ** سلوك ** **”**	“*Suluk*”	Behavior	12 (10.44%)
**“** ** عدواني ** **”**	“*Audwani*”	Aggressive	9 (7.83%)
**“** ** متكرر ** **”**	“*Mutakarir*”	Repeated	7 (5.22%)
**“** ** سلبي ** **”**	“*Salbi*”	Negative	5 (4.35%)
**“** ** لفظي ** **”**	“*Lafzi*”	Verbal	5 (4.35%)
**“** ** فعل ** **”**	“*Fi’el*”	An Act	4 (3.48%)
**“** ** الأذى ** **”**	“*Al Az’a*”	Harm	3 (2.61%)
**“** ** الإساءة ** **”**	“*Al Eisa’ah*”	Offence	3 (2.61%)
**“** ** العنف ** **”**	“*Al Onf*”	Violence	3 (2.61%)
**“** ** قول ** **”**	“*Qawl*”	Saying	3 (2.61%)
**“** ** الإعتداء ** **”**	“*Al Ei’tida*”	Assault	2 (1.74%)
**“** ** النفسي ** **”**	“*Al nafsi*”	Psychological	2 (1.74%)
**“** ** تعمد ** **”**	“*Ta’ammod*”	Deliberate	2 (1.74%)
**“** ** جسدي ** **”**	“*Jasadi*”	Physical	2 (1.74%)
**“****غير** **سوي** **”**	“*Gayr Sawi*”	Misbehavior	2 (1.74%)
**“** ** مؤذي ** **”**	“*Mo’zi*”	Harmful	2 (1.74%)
**“** ** متعمد ** **”**	“*Mota’ammad*”	Intentional	2 (1.74%)

## Data Availability

All data generated or analyzed during this study are included in this submission.

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
