# Peer review of "The Perception and Roles of School Mental Health Professionals Regarding School Bullying (Suluk Audwani) in Oman: A Qualitative Study in an Urban Setting"

_ijerph, 2024, doi:10.3390/ijerph21080991_

Round 1
Reviewer 1 Report
Comments and Suggestions for Authors
Dear Authors,
I have reviewed your manuscript entitled "The perception and roles of school mental health professionals towards school bullying (Suluk Audwani) in Oman: A qualitative study in an urban setting" and would like to congratulate you on your work.
I consider the paper to be quite comprehensive. However, I leave below some details that would be appropriate to modify:
- When you mention table 1 for the first time in the Participants section the table should appear below. You can solve this in two ways: either do not mention table 1 in that section (just say that there were 50 participants) or, on the contrary, add all the information about demographic data in that section and then insert the table.
- Table 2 where it is placed makes no sense. I recommend that you include it in the section where it is mentioned (2.4 Procedure).
- Explain in more detail how many focus groups emerged and how you grouped the participants in those focus groups. Why were they done in two sessions? Were they both on the same day?
- In the conclusions section, it would be interesting to add future proposals related to your work, as well as the practical implications of your research within the scientific evidence.
These would be the suggested recommendations. Success in your research.
Author Response
We would like to extend our sincere gratitude for your thorough review and constructive comments on our manuscript entitled "The Perception and Roles of School Mental Health Professionals Toward School Bullying (Suluk Audwani) in Oman: A Qualitative Study in an Urban Setting." Your insights have been invaluable in improving the quality of our work. Below, using the point-counterpoint format, we address each of your suggestions and explain how we have made it to the manuscript. All modifications have been incorporated into the text and are shown as tracked changes.

Reviewer 2 Report
Comments and Suggestions for Authors
This is a well-written study investigating perceptions and roles of school mental health professionals toward school bullying in Oman.
I have a few comments
1: The authors should clarify the recruitment process of mental health professionals and the participation acceptance rate.
2: The authors may consider adding a section comparing the perceptions of mental health professionals to their counterparts in Western countries regarding school bullying. This could provide a better understanding of the cultural perspective of forming such perceptions.
3: References 6, 7, 8, 10, 17, 22, 25, 26, 36, 37, 39, 41, 53 need to be revised. I could not reach most of them. Links should be added. The references should be formatted per the requirement of the journal.
Author Response

(The authors gave the same response as above.)

Round 2
Reviewer 1 Report
Comments and Suggestions for Authors
Dear Authors,
I have reviewed your manuscript again and have noted that you have carried out all the recommendations included in the first review. For my part, the article is now complete and ready for publication.
Congratulations on your work.
Best regards.